

# Near wake analysis of actuator line method immersed in turbulent flow using large-eddy simulations

Jörn Nathan[1], Christian Masson[1], and Louis Dufresne[1]

[1]ÉTS, Univ. du Québec, Mechanical Engineering, Montréal H3C 1K3, Canada

*Correspondence to:* Jörn Nathan (joern.nathan.1@ens.etsmtl.ca)

**Abstract.** The interaction between wind turbines through their wakes is an important aspect of the conception and operation of a wind farm. Wakes are characterized by an elevated turbulence level and a noticeable velocity deficit which causes a decrease in energy output and fatigue on downstream turbines. In order to gain a better understanding of this phenomenon this works uses large-eddy simulations together with an actuator line model and different ambient turbulences imposed as boundary conditions. This is achieved by using the SOWFA framework from NREL (USA) which is first validated against another popular CFD framework for wind energy, EllipSys3D, and then verified against the experimental results from the MEXICO and NEW MEXICO wind tunnel experiments. By using the predicted torque as a global indicator, the optimal width of the distribution kernel for the actuator line is determined for different grid resolutions. Then the rotor is immersed in homogeneous isotropic turbulence and a shear layer turbulence with different turbulence intensities, allowing to determine how far downstream the effect of the distinct blades is discernible. This can be used as an indicator for the extents of the near wake for different flow conditions.

## 1 Introduction

An important aspect for the conception of wind farms is the turbine spacing which depends on the interaction of wind turbines through their wakes. This phenomenon can decrease the wind park energy output by up to $20\%$ due to the velocity deficit propagated by the wakes (Manwell et al., 2010). Additionally, it can increase the turbine fatigue due to the increased turbulence intensity. In order to study wake interactions, the flow around the rotor has to be modelled correctly. Hence the model should account for the apparition of turbulent structures of different magnitudes. For instance the vortices created by the blade tips and its interaction with the ambient turbulence.

As opposed to the far wake region (Olivares Espinosa, 2017), the near wake representation in a computational fluid dynamics simulation depends heavily on the applied rotor model. Approaches range from an actuator force representation inserted as momentum sink in the Navier-Stokes equations to full rotor modelling where the attached boundary layers on the blades are simulated (Sanderse et al., 2011). This work will apply the actuator line method (ALM) in order to model the transient behaviour of the rotor by representing distinctly the rotating blades as presented by Troldborg (2009). Each blade is represented by a force line allowing to reproduce the helicoidal vortical structure in the near wake allowing to to assess its interaction with the flow.



In order to evaluate the soundness of the present method a comparative study of the SOWFA framework, from NREL, and EllipSys3D, from DTU, was conducted as initally presented in Nathan et al. (2017). Based on this study, the method used throughout this work will be evaluated before proceeding to establish the base case for the non-turbulent inflow. For establishing the base case, the optimal width of the distribution kernel of the forces of the actuator line is determined. While previous

work often focused on numerical stability as in Troldborg (2009) or Ivanell et al. (2010), when choosing the distribution width, Martínez-Tossas et al. (2015a) states that with decreasing distribution width, the line forces are getting too concentrated resulting in a wrong prediction of the rotor torque. Hence this work tries to evaluate the optimal width for each mesh resolution by using the predicted torque as a global indicator.

For the introduction of a turbulent inflow different methods exist for imposing a stastically generated velocity field, such as

inserting it via a momentum sink as done in Troldborg et al. (2011) or as boundary conditions as done in Olivares Espinosa (2017). This work adheres to the latter approach, as it was seen as more straightforward than the conversion of the velocity field to a force which then is translated back to the velocity field by the numerical solver as done the former approach.

Then shear layer turbulence (Muller et al., 2014) is introduced exposing the rotor model to a more realistic wind flow situation bearing more resemblance to applied wind energy. This novel approach takes into consideration the temporal evolution of the

sheared velocity field, hence allowing it to be imposed as boundary condition as well. Also this method was not yet applied to the actuator line method.

Finally, the numerical results are used to examine the spatial extents of the near-wake region. While in previous work such as Krogstad and Eriksen (2013) or Sarmast et al. (2016) often the profiles of velocity deficits or turbulent kinetic energy are taken into consideration for evaluating the near-wake, this work uses the energy spectra to determine how far downstream the

discernible effects of the distinct blades are noticable. While the analysis of the turbulent inflow in previous work (Olivares Espinosa, 2017) has often been conducted using energy spectra, seldom energy spectra including the rotor effects are included in the analysis of the near-wake. As with increasing turbulence intensity the statistical convergence tends to be longer, the energy spectra approach in this work permits the analysis of the spatial extensions of the near-wake region even without fully convergence of second-order statistics.

## 2   Numerical methodology


The numerical simulations are based on the incompressible Navier-Stokes equations

$$\frac{\partial \mathbf{U}}{\partial t} + \nabla \cdot (\mathbf{U}\mathbf{U}) - \nabla \cdot (\nu \nabla \mathbf{U}) = -\nabla p/\rho + \mathbf{F} \qquad (1)$$

$$\nabla \cdot \mathbf{U} = 0 \qquad (2)$$

with $\mathbf{F}$ representing the actuator force inserted as a momentum sink, $\mathbf{U}$ as the velocity, $\nabla p/\rho$ as the modified pressure and $\nu$ as

the kinematic viscosity. In  section 2.1 the rotor model and the derivation of the force term $\mathbf{F}$ are discussed. Then in  section 2.2 an overview is given on the methods generating the two different ambient turbulences and how they are imposed as boundary conditions. Finally there is a summary of the numerical framework and its setup in  section 2.3.





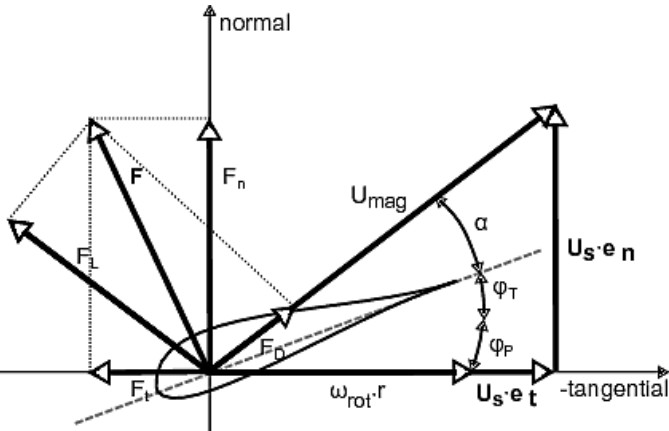

**Figure 1.** Geometry and forces in an airfoil section of the blade.

## 2.1 Rotor model

The force term $\mathbf{F}$ in Eqn. (1) is obtained using

$$\mathbf{F} = \mathcal{G} * \left( f_{tip} \cdot (F_L \mathbf{e}_L + F_D \mathbf{e}_D) \right) \tag{3}$$

with the lift and drag forces shown in Figure (1) and defined as

$$F_L = \frac{1}{2} c_l \, U_{mag}^2 \, c \, l_s \tag{4}$$

$$F_D = \frac{1}{2} c_d \, U_{mag}^2 \, c \, l_s \tag{5}$$

with $c_l$ and $c_d$ as the lift and drag coefficient, $U_{mag}$ the sampled velocity magnitude in the blade reference frame, $c$ the chord width, $l_s$ the length of the actuator segment, Gaussian kernel $\mathcal{G}$ and tip correction $\left( f_{tip} \right.$.

The airfoil coefficients where selected among the ones presented in Schepers et al. (2012) based on the blade Reynolds number $Re = U_{mag} \, c/\nu$ with $c$ as chord and $\nu$ as the kinematic viscosity. This data was obtained from wind tunnel experiments and therefore it does not include the stall delay due to boundary layer stabilizing effects such as Coriolis and centrifugal forcing which enhance the lift of the airfoil (Vermeer et al., 2003). In the case of the MEXICO rotor an adaption was proposed by Shen et al. (2012) circumventing this issue, but at the same time proposing a solution tuned for a known outcome. Hence this works adopts the unmodified 2D airfoil data. Then the forces in Eqn. (4) and Eqn. (5) are projected in the blade reference frame using the unit vectors $\mathbf{e}_L$ and $\mathbf{e}_D$ in Eqn. (3).

While the Glauert tip correction $f_{tip}$ was originally intended (Glauert, 1935) to represent the otherwise absent tip vortices in the actuator disk model, it still proves advantageous for the ALM at lower resolutions as shown in Nathan (2018). Due to the relatively low resolution, the shed tip vortices from an ALM are much larger than the ones observed experimentally. Hence the induction caused by the simulated vortices is weaker than in reality and the Glauert tip correction permits to compensate for it (Nathan et al., 2017).



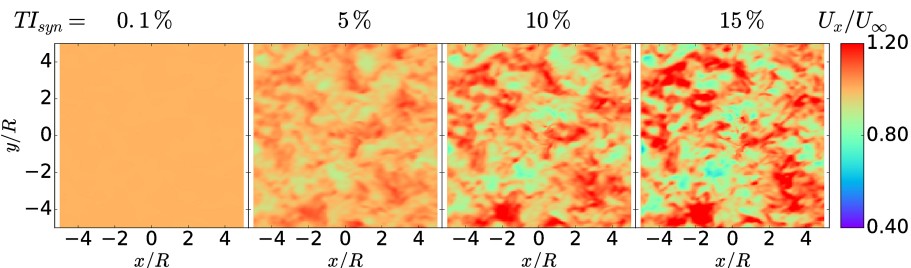

**Figure 2.** Midplane at $y/R = 0$ of instantaneous normalized axial velocity component $U_x/U_\infty$ showing homogeneous isotropic turbulence for different turbulent intensities $TI_{syn}$ in numerical domain.

Finally in order to avoid spurious oscillations around the point of the inserted force, the punctual force is distributed using a kernel $\mathcal{G} * (\cdot)$. As done in previous works such as Troldborg (2009) or Olivares Espinosa (2017) this work adheres to a normal distribution with the distribution width $\epsilon = \sigma \Delta x$.

## 2.2 Turbulence inflow generation

### 2.2.1 Homogeneous isotropic turbulence

A synthetic velocity field representing homogeneous isotropic turbulence based on the von-Kármán energy spectrum (Pope, 2000)

$$E(k) = \alpha \epsilon^{2/3} L^{5/3} \frac{L^4 \kappa^4}{(1 + L^2 \kappa^2)^{17/6}} \tag{6}$$

is obtained by using the algorithm proposed by Mann (1998). The technical details can be found in the article of Mann (1998)

or more recently in Olivares Espinosa (2017). The main parameters for this approach are the integral length-scale $L$ and the coefficient $\alpha \epsilon^{2/3}$ which can be used as a scaling factor to obtain the desired amplitude of the turbulent structures. The range of wavenumber $\kappa$ depends on the grid resolution and dimension extents. Hence, these parameters determine the ability of the numerical mesh to resolve a certain range of turbulent scales.

While several implementations of this method exist e.g. Olivares Espinosa (2017) or Muller et al. (2014), the implementation

of Gilling (2009) was chosen for synthesizing the HIT for several reasons. It is in public domain, it corrects for a divergence free velocity field and allows to impose HIT at the boundaries at relatively low computational cost.

In Figure (2) the midplane of a generated turbulent field is shown for different turbulence intensities. The flow structures are identical apart from the different scaling of the velocity fluctuations. This results from using the same seed for the random number generator in the Mann algorithm and by scaling the obtained velocity field with $\alpha \epsilon^{2/3}$ to obtain the desired synthetic

turbulence intensity $TI_{syn}$.

Contrary to Troldborg (2009) where the synthetic turbulence is imposed as a momentum source, in this work it is imposed as a boundary condition as done in Muller et al. (2014) and Olivares Espinosa (2017). The velocities are imposed by convecting the velocity field of the synthetic turbulence through the computational domain by the mean velocity $U_\infty$ at each simulation

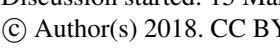



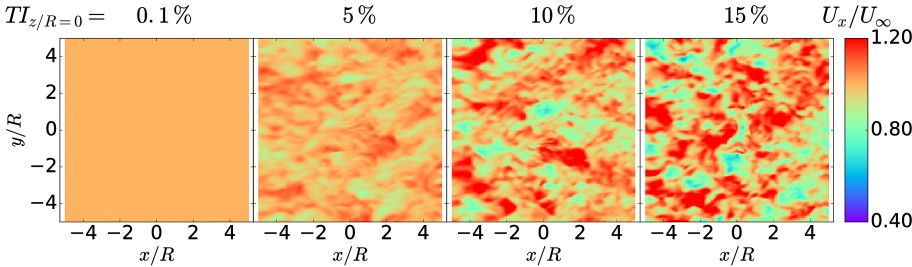

**Figure 3.** Horizontal plane of instantaneous axial velocity component $U_x$ in sheared flow for different longitudinal turbulence intensities $TI_{z/R=0}$ at hub position.

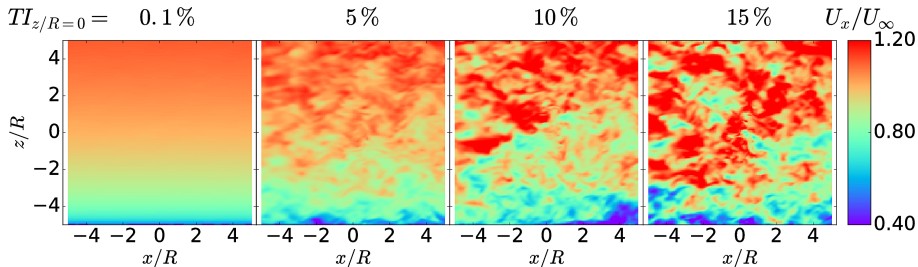

**Figure 4.** Vertical plane of instantaneous axial velocity component $U_x$ in sheared flow for different longitudinal turbulence intensities $TI_{z/R=0}$ at hub position.

time step. They are then projected by trilinear interpolation onto the computational points. In order to speed up the statistical convergence, the simulation is also initialized with the synthetic turbulence field.

### 2.2.2 Shear layer turbulence

Based on the Mann algorithm (Mann, 1998), Muller et al. (2014) developed a method to impose the synthetic turbulence on a sheared flow as boundary condition, including the evolution of the vortical structures. A typical flow field generated by this algorithm can be seen in Figure (3) and Figure (4).

The mean velocity profile is obtained via the power law

$$U_x = U_{ref}\left(\frac{z}{z_{ref}}\right)^{\alpha_p} \tag{7}$$

hence the velocity at the bottom of the domain has not necessarily to be zero. Therefore the computational mesh can be much smaller than in a wall-resolved flow, as its mesh has to include the ground and has to have a high refinement in this region. The reference height $z_{ref}$ was set at hub height and the reference velocity was set at $U_{ref} = 15\,m/s$. The parameter $\alpha_p$ can usually be deduced from experimental measurements if available. As they were not available for this experiment, standard conditions are assumed with $\alpha_p = 1/7$ (Pope, 2000).





## 2.3 Numerical framework

This work is realized within the open-source framework OpenFOAM[1] (version 2.2.2) together with the SOWFA[2] project, which contains a similar implementation of the ALM as presented by Troldborg (2009). A more detailed explanation of the implementation can be found in Martínez-Tossas et al. (2016). OpenFOAM is a set of libraries and executables entirely written

in C++. While the first released scientific article about the framework was by Weller et al. (1998) its inner workings are described more in-depth by Jasak (1996).

The computational domain is cubic with an edge length of $L_x = L_y = L_z = 10\,R$ with $R$ as the rotor radius and the rotor positioned at the domain center. The cells in the rotor vicinity are refined in the range of $-0.4 \leq x/R \leq 0.4$ with the size $\Delta x = D/128$. Within SOWFA several refinement zones are applied each time halving the cell edge length as also done in Vanella

et al. (2008). The final mesh size consists of $1.9 \cdot 10^6$ cells. The technique used in SOWFA proves highly advantageous in terms of computational cost and its impact on the results are examined in a sensitivity study in Nathan (2018).

For the boundary conditions the velocity is imposed as uniform inflow velocity of $\mathbf{U} = (U_\infty, 0, 0)$ for the non-turbulent flow and in the turbulent cases as the synthetic velocity as explained in section 2.2. The lateral boundaries are set as symmetric for the non-turbulent and homogeneous isotropic turbulence case. For the shear layer turbulence the velocity is also imposed at the

lateral boundaries.

The large eddy simulations use the dynamic Lagrangian sub-grid scale model (Meneveau et al., 1996). For the discretization of the convective term a linear combination of $75\%$ central differencing and $25\%$ of a second-order upwind scheme is applied as presented by Warming and Beam (1976). In OpenFOAM terminology this scheme is called "Linear-upwind stabilized transport" (LUST). The choice of the scheme is made as a trade-off between the accuracy of a linear discretization and the

stability of an up-winding scheme. This scheme proved to preserve well the turbulent structures (Nathan, 2018). The remaining spatial terms are discretized by central differencing and for the time discretization the Crank-Nicolson method is used.

The pressure is resolved using a geometric agglomerated algebraic multi-grid solver and the remaining variables are solved for with a bi-conjugate gradient method using a diagonal-based incomplete LU preconditioner. The total simulation run-time comprises 60 rotor revolutions and the time step has to be small enough to avoid the actuator point representing the blade tip

skipping a computational cell during rotation. It is also an integer fraction of the rotor revolution time. This run-time is chosen as first and second order statistics are deemed to be converged.

For the parametrization of the ALM, different distribution widths are chosen in order to obtain the optimum for the examined case and 40 actuator points are used to represent one blade in accordance with what was found in Nathan (2018).

---

[1]OPENFOAM® (Open source Field Operation And Manipulation) is a registered trade mark of OpenCFD Limited, producer and distributor of the OpenFOAM software via wwww.openfoam.com.

[2]NWTC Design Codes (SOWFA (Simulator fOr Wind Farm Applications) by Matt Churchfield and Sang Lee) http://wind.nrel.gov/designcodes/simulators/SOWFA/. NWTC (National Wind Technology Center) is part of NREL (National Renewable Energy Laboratory) based in Golden, CO, USA.







**Figure 5.** Axial profiles of phase averaged (rotor position $\Psi = 0^{\text{o}}$) velocity components for different flow cases an inboard and an outboard radial position.

## 3 Results

### 3.1 Validation and verification

The implementation was validated against EllipSys3D and verified against the MEXICO and NEW MEXICO experiment in Nathan et al. (2017). A comparison of the axial profiles of the velocity components can be seen Figure (5) showing that 5 both codes reproduce very similar results in the near wake further away from the rotor. While the axial induction for $x/R \geq$ 0.5 corresponds very well to the experimental data, the velocity deficit is under-predicted in the immediate rotor vicinity $-0.5 \leq x/R \leq 0.5$. For the high velocity case ($U_\infty = 24 \, m/s$) the vortex sheets shed from the blades become visible by the oscillations in the axial velocity component $U_x$.





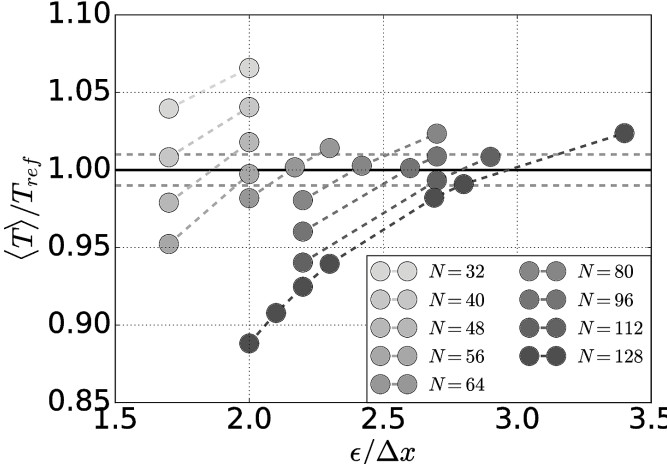

**Figure 6.** Relation between $\epsilon/\Delta x$ and resulting global torque normalized by reference torque for $U_\infty = 15\,m/s$.

## 3.2 Non-turbulent flow

When refining the grid using the actuator line method the distribution parameter $\epsilon$ has to be adjusted to obtain a global torque $\langle T \rangle$ close to the reference value $T_{ref}$. In the following only the case for $U_\infty = 15\,m/s$ will be examined as the other cases in Nathan et al. (2017) served as extreme cases for determining how the model behaves at its limits.

Instead of relying on a constant $\epsilon/\Delta x$ for different grid resolutions, this work adapts $\epsilon/\Delta x$ depending on the grid resolution, or number of cells across the rotor diameter $N = 2R/\Delta x$. This can be seen as a first step towards an actuator surface method, where the force is distributed with respect to the blades chord. The results are shown in Figure (6). A confidence interval of $\pm 1\,\%$ was established around the reference torque value $T_{ref}$. Through iterations, an optimal distribution parameter is found to fall in this range.

The lower bound for the distribution parameter here is $\epsilon = 1.7\Delta x$ for the sake of numerical stability of the here chosen method. Other frameworks applying a different numerical discretization can go even lower e.g. in Ivanell et al. (2010). By doing so it can be seen that the best solution in terms of global torque for a resolution of $N = D/\Delta x = 32$ is off by around $4\,\%$ in Figure (6).

    As a general trend it can be seen that $\epsilon/\Delta x$ has to be increased with increasing resolution. This stems from the fact that

by refining the mesh with a constant $\epsilon/\Delta x$ the punctual induction caused by the blade would be too high and eventually the torque would be below the reference value, e.g. for $\epsilon = 2\Delta x$ for $N \geq 64$. On the contrary, when having a very low resolution a constant $\epsilon$ distributes the force too widely, causing a lower induction around the rotor resulting in an overestimation of the torque, e.g. for $\epsilon = 2\Delta x$ for $N \leq 48$.

    The optimal distribution parameter $\epsilon/\Delta x$ found in Figure (6) are now shown in dependence of the grid resolution $N =$

$D/\Delta x$ in Figure (7). It seems as if this value would reach eventually an asymptotic limit for higher resolutions. When looking at a more theoretical approach in Martínez-Tossas et al. (2015b) it is suggested that the optimal distribution width $\epsilon$ lies between




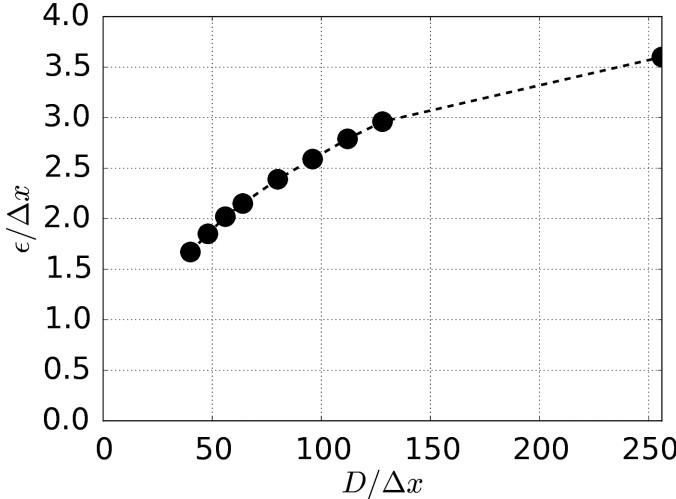

**Figure 7.** Optimal $\epsilon/\Delta x$ over number of cells for resolving one rotor diameter $D/\Delta x$.

$0.14 - 0.25$ of the chord $c$ whereas in this case for $D/\Delta x = 128$ the $\epsilon/c$ lies between $0.5 - 8.9$ depending on the spanwise location. The observation made by Martínez-Tossas et al. (2015b) is backed by Shives and Crawford (2013) where $\epsilon/c$ falls in the same range. But it should be kept in mind that Shives and Crawford (2013) use a much higher grid resolution allowing $\epsilon/\Delta x \geq 4$ and in the case of Martínez-Tossas et al. (2015b) even $\epsilon/\Delta x \geq 5$.

The curvature in Figure (7) also confirms the findings of Jha et al. (2013) that keeping the relation $\epsilon/\Delta x = const$ while increasing the resolution is not a very good solution. While Ivanell et al. (2010) suggests to choose the smallest possible distribution width $\epsilon$ in order to minimize interactions with the vortical structures ($\epsilon/\Delta x = 1$), the observations made here fall more in line with work such as Shives and Crawford (2013) suggesting $\epsilon/\Delta x$ to be adapted to the physical model and in order to distribute the force over a meaningful length scale.

For an excerpt of the resolutions presented in Figure (6) the radial profiles of the velocity components can be found in Figure (8). It can be seen that the method seems to converge towards a solution when refining the mesh. As shown in Figure (6) the lowest resolution at $N = 32$ over-predicts the torque by distributing the force to widely which also reflects in the low axial induction downstream at $x/R = 0.13$. Despite following well the trend of the experimental values the method seems to converge towards radial profiles which are especially off in the tip and hub region where the strongest vortices are shed. These

are limitations intrinsic of the ALM which is less apparent when using high fidelity approaches such as full rotor simulations (Carrión et al., 2015).

In Figure (9) the shed vortical structures can be seen in dependence on the grid resolution. While the root vortex is rather diffuse, a clear tip vortex can be noticed. It is interesting to notice the vortices shed around mid-span due to sub-optimal choice of the airfoils of blade causing a sudden change in circulation.

In order to estimate the resolution necessary to obtain tip vortex radii as seen in the MEXICO experiment the vortex radii are shown in Figure (10). The vortex radius $r_{core}$ is defined as the limit containing $99\%$ of the circulation. The Gaussian





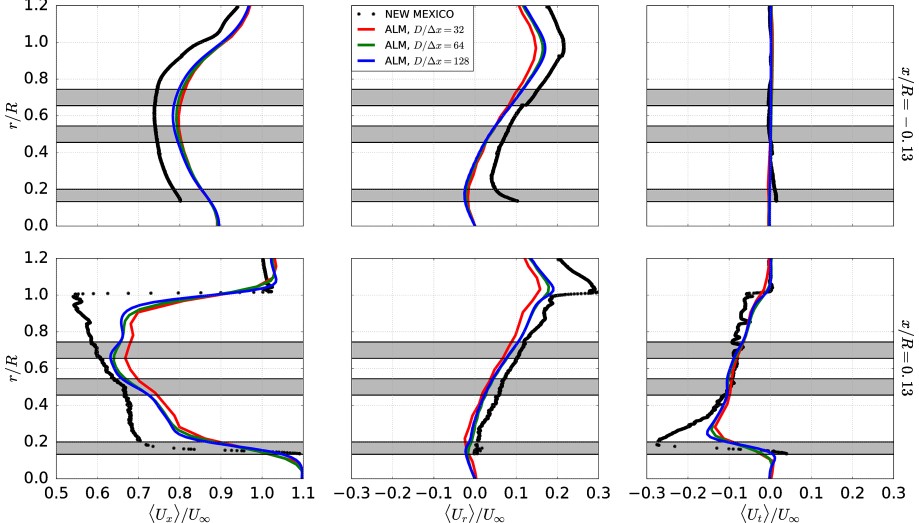

**Figure 8.** Radial profiles of time-averaged velocity components $\langle U_x \rangle$ (axial), $\langle U_r \rangle$ (radial) and $\langle U_t \rangle$ (tangential) for ALM in different grid resolutions at $x/R = 0.13$.

distribution is used as an approximation for the vorticity distribution within the vortex. This assumption is normally applied for low Reynolds number flows, while this case exhibits a Reynolds number of $Re = \Gamma/\nu = 220 \cdot 10^3$. Nevertheless this approximation is used in order to be able to draw an analogy between the experimental and the numerical results. It holds fairly well when comparing the Gaussian distribution and the vorticity for $N = 128$ as shown in Nathan (2018). Hence by assuming a Gaussian distribution for the vortices in the MEXICO experiment, a corresponding distribution parameter $\epsilon$ can be deduced as shown in Figure (10).

This would necessitate a resolution of $N \geq 4096$ for the here presented case which would result in a computational grid beyond any justifiable computational scope. Full rotor calculations as conducted by Carrión et al. (2015) allowed to obtain tip vortices of $r_{core}/R \approx 0.012$ for $N \approx 900$ in the tip region which corresponds very well to results in Figure (10). Another result for the vortex radius can be found in Nilsson et al. (2015) where for $\epsilon/\Delta x = 1$ and $N \approx 244$ in the tip region a vortex core radius of $r_{core}/R \approx 0.055$ was found. Despite the radii in this work and the references are calculated based on three different methods, the results fall within the same range.

## 3.3 Homogeneous isotropic turbulence

In Figure (11) the longitudinal evolution of the turbulence intensities can be seen. There is a stronger decay for higher turbulence intensities which was also found in Olivares Espinosa (2017). In that work EllipSys3D was compared to a solution based on OpenFOAM and it was found that over the same longitudinal distance of $10R$ a decay of $48\%$ and $44\%$ occurred for each framework respectively. This stands in a stark contrast to the $4\%$ in this case for the high turbulence intensity case. This huge

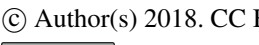



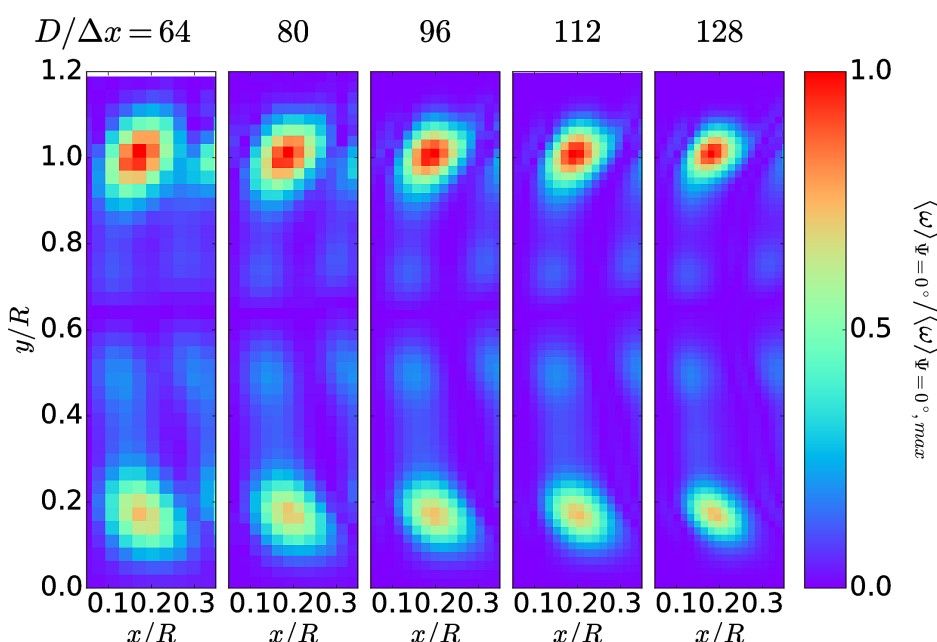

**Figure 9.** Normalized vorticity $\langle\omega\rangle_{\Psi=0^\circ}/\langle\omega\rangle_{\Psi=0^\circ,max}$ in the near wake for different grid resolutions.

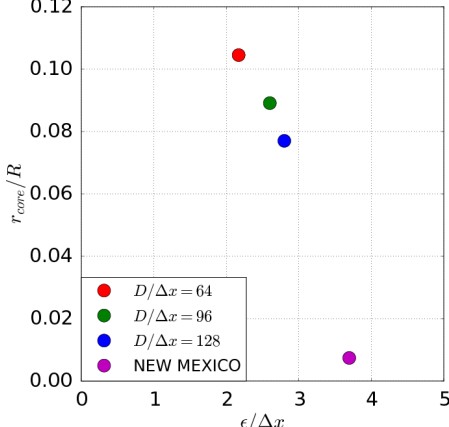

**Figure 10.** Normalized vortex radius $r_{core}/R$ over normalized distribution parameter $\epsilon/R$.





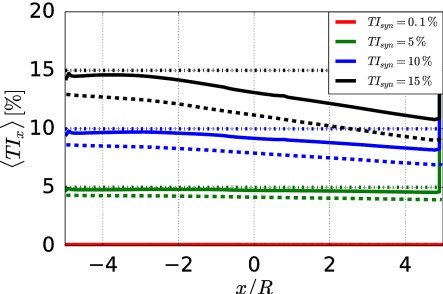

**Figure 11.** $\langle TI \rangle$ with $\langle TI_{res} \rangle$ (dashed line), $\langle TI_{res} \rangle + \langle TI_{sgs} \rangle$ (solid line) and $TI_{syn}$ (dotted line) as reference.

decay, which is even more significant for EllipSys3D, necessitates to approach the introduction of the turbulence close to the turbine for high turbulence intensity cases (Olivares Espinosa, 2017).

An important aspect when imposing a synthetic turbulence as boundary conditions of a CFD simulation is respecting the Nyquist–Shannon sampling theorem (Shannon, 1949) as also mentioned by Muller et al. (2014). Hence a study considering

different ratios between the grid resolution of the synthetic turbulence and the simulation was undertaken. It is found that the higher the computational resolution is compared to the one of the synthetic turbulence, the less the turbulence intensity decays in longitudinal direction. While the criterion of Nyquist–Shannon states that the resolution of the computational domain should be at least twice as big, this work uses the ratio of $dx/\Delta x = 2.5$ with $dx$ as the cell width of the synthetical field and $\Delta x$ as the cell width of the computational mesh.

When taking the case for $TI_{syn} = 5\%$ it is interesting to notice that while the resolved $TI$ (green dashed line) is around $4.2\%$ at the rotor position $x/R = 0$ a huge part of the difference in relation to the imposed turbulence falls in the SGS model with $\langle TI_{res} \rangle + \langle TI_{sgs} \rangle = 4.8\%$ and finally just a relatively small amount of the turbulent intensity or turbulent kinetic energy is "lost" by numerical dissipation.

Despite the fact that the computational grid respects the Nyquist–Shannon criterion for signal sampling in respect to the

synthetic grid, immediately at the inlet a part of the turbulence falls in the sub-grid range. Due to the numerical dissipation caused by the differencing schemes and turbulence modelling the energy cascade hands down its energy to lesser scales than the resolved ones.

It should be kept in mind, that the turbulence intensity the rotor model is experiencing through velocity sampling is the resolved turbulence intensity $TI_{res}$ and the sub-grid turbulence intensity $TI_{sgs}$ is therefore only felt indirectly, by an augmen-

tation of the effective viscosity. When looking at the fraction of the resolved turbulent kinetic energy over the total turbulent kinetic energy, it can be seen that the resolved scales exceed $96\%$ which lies well above the criterion of $80\%$ proposed by Pope (2004).

In Figure (12) the effects of the ambient turbulence on the turbine wake are shown. While there are no noticable impacts for the low turbulence case with $TI_{syn} = 0.1\%$ the beginning of strong non-linear interactions can be observed for $TI_{syn} > 0.1\%$.

For $TI_{syn} = 15\%$ the inflow turbulent structures seem to outgrow the structures created by the wind turbine.




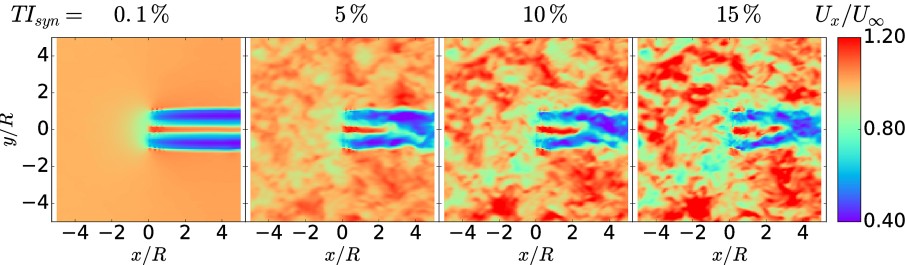

**Figure 12.** Instantaneous axial velocity component $U_x$ of wind turbine wake immersed in HIT.

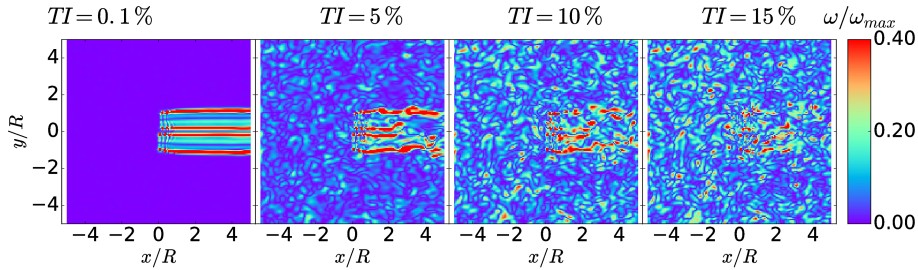

**Figure 13.** Instantaneous vorticity fields for different turbulence intensities.

In Figure (13) it can be seen how the strength of the vortical structures of the ambient fluid increases with higher turbulence intensity up-to the point for $T_{syn} = 15\%$ where its amplitude equals almost the one emitted by the rotor model. In Figure (14) the impact of the rotor presence on the energy spectrum can be seen. The wavenumber $\kappa_p$ relating to the frequency of a blade passage (three times rotor frequency) obtained by $\kappa_p = 2\pi 3 f / U_\infty$ shows a very distinct peak and its higher harmonics at the multiples of $\kappa_p$. As the velocity time series obtained from the simulations do not exhibit periodicity, the Welch method (Welch, 1967) is used to generate the energy spectra.

It is interesting to notice the distinct peaks in the spectra occur at the wavenumber relating to the frequency of the blade passage and its harmonics. The harmonics are caused by the strong excitement of the fluid by the blade passage and its interaction with the non-linear term in the NS equations. As the blade forces and hence the strength of the tip vortices are very comparable, the peaks are very similar among the different cases for $-0.4 \leq x/R \leq 0.2$. The higher the turbulent kinetic energy content stemming from the ambient flow the faster the peaks are dampened and blend into the ambient flow. For example there is almost no discernible effect by the blade at $x/R = 0.4$ for $TI_{syn} = 15\%$ while for $TI_{syn} = 0.1\%$ the velocity oscillations are still very noticable. Although it is of lesser amplitude also the upstream region is under the influence of the distinct blades up-to a certain extent.

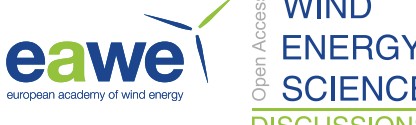


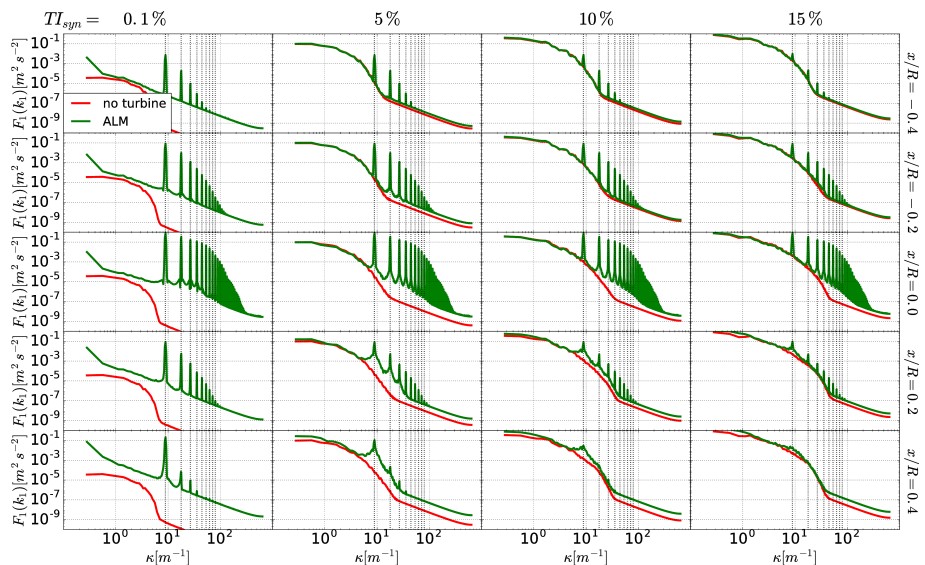

**Figure 14.** Impact of rotor presence on energy spectrum with wavenumber relating to 3 times rotor frequency and its higher harmonics indicated by dotted black lines. Each column represents the different turbulence intensity cases and the rows are representing different axial positions of the spectra.

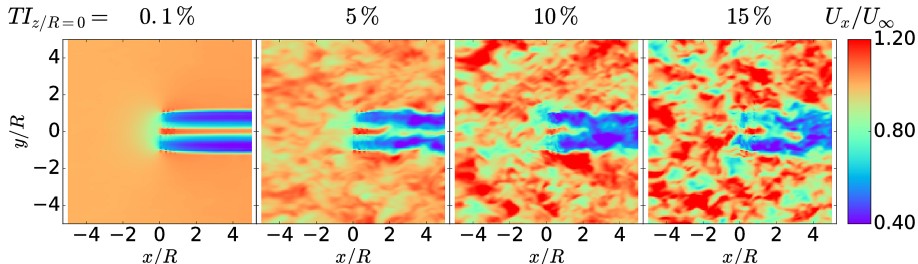

**Figure 15.** Horizontal plane of instantaneous axial velocity component $U_x$ of wind turbine wake immersed in shear turbulent flow at hub height.

## 3.4 Shear layer turbulence

The instantaneous and mean velocity fields with an immersed rotor can be seen in Figure (15). The horizontal plane at hub height again seems to behave similar to the HIT cases in Figure (12). When looking at the vertical planes in Figure (16) the influence of the sheared flow can be seen by a higher velocity deficit in the wake on the lower half of the rotor.

5    When looking at the instantaneous normalized vorticity in Figure (17) it can be seen that while the vortical structures emitted by the rotor are prevalent for low turbulence intensity cases, they seem to get even for $TI_{x,z/R=0} \geq 10\,\%$. Hence the turbulent structures of the ambient flow become as significant as the ones emitted from the wind turbine. This is of particular interest for estimating the range up-to which the effects of the distinct blades can be felt which serves often as a measurement for





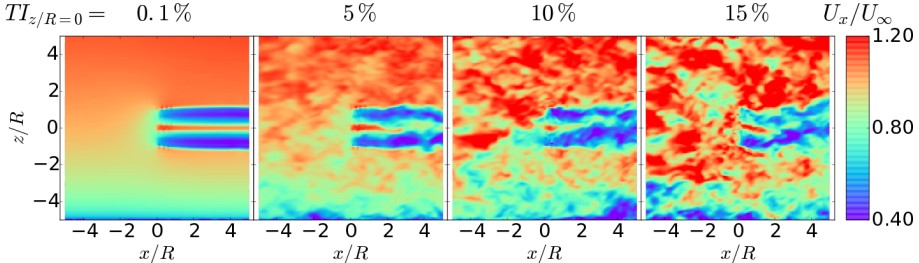

**Figure 16.** Vertical plane of instantaneous axial velocity component $U_x$ of wind turbine wake immersed in shear turbulent flow.

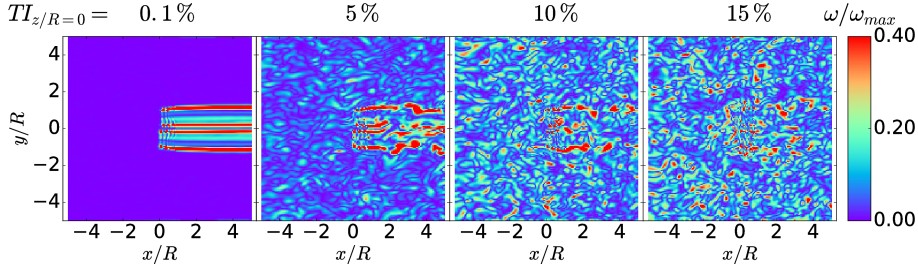

**Figure 17.** Instantaneous normalized vorticity $\omega/\omega_{max}$ in sheared flow for different longitudinal turbulence intensities $TI_x$ at hub position. horizontal plane at $z/R = 0$

determining the near wake. In the vertical plane in Figure (18) it can be seen that there is an increase in the vorticity magnitude towards the ground.

Looking at the energy spectra in Figure (19) reveal a similar picture as shown above for the case of homogeneous isotropic turbulence. As the blade forces and hence the strength of the tip vortices are very comparable, the peaks are very similar among

5    the different cases for $-0.4 \leq x/R \leq 0.2$. Due to the dissipation caused by the ambient turbulence these peaks dampen at a different pace as seen at $x/R = 0.4$.

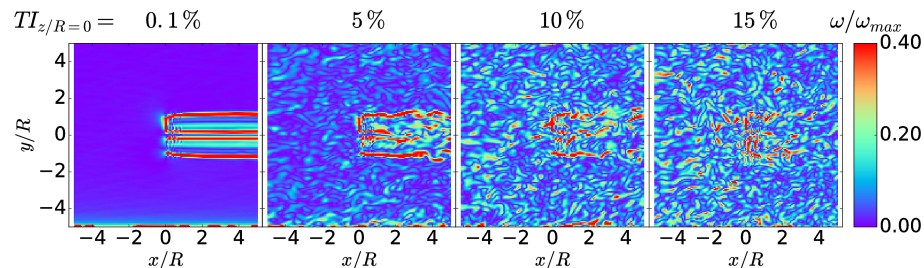

**Figure 18.** Instantaneous normalized vorticity $\omega/\omega_{max}$ in sheared flow for different longitudinal turbulence intensities $TI_x$ at hub position. vertical plane at $y/R = 0$



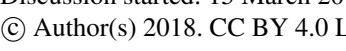
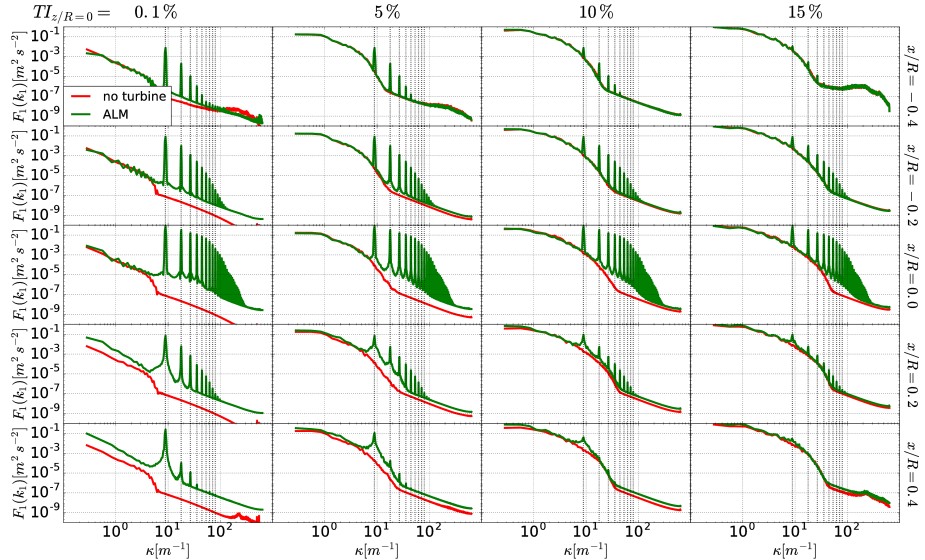

**Figure 19.** Spectra of different axial positions at hub height $z/R$ for the shear layer turbulence case. Each column represents the different turbulence intensity cases and the rows are representing different axial positions of the spectra.

While before and at the rotor position for $-0.4 \leq x/R \leq 0$ the peaks remain very distinct, vortical structures by the ambient fluid and emitted by the blade cause the injected peaks to dampen and distributing energy to adjacent wavenumbers as seen clearly for $x/R \geq 0.2$. Depending on the level of the ambient turbulence the peak gets attenuated up-to a point where it blends almost completely in with ambient turbulence as seen for $TI_{x,z/R=0} = 15\%$ at $x/R = 0.4$. This relates to the observation

made earlier when looking at Figure (15) and Figure (17) where the ambient structures are almost as important as the structures emitted by the blade.

This is particularly interesting for examining the reach of the here used rotor model and the distinct presence of the separate blade forces. It seems that for a realistic case with a turbulent shear flow and a turbulence intensity of $TI_{x,z/R=0} \geq 10\%$ the velocity fluctuations at $x/R = 0.4$ already seem to have only a weak relation to the injected turbulence by the rotor but a much

stronger one to the ambient turbulence.

Although flow properties were maintained at a similar level, the sheared flow has a clear impact on the power extraction and also on vortex properties of the structures emitted by the blade. A very interesting observation is the fact that for higher turbulence intensities the effects of the distinct blades using the ALM seems to vanish at relatively short downstream distances. This poses the question of the usability of the ALM when arguing for its capabilities of representing the transient behaviour

and its impact on downstream turbines.



## 4   Conclusions

By using a validated actuator line implementation (Nathan et al., 2017), it was shown that the distribution width $\epsilon/\Delta x$ has a non-linear dependence on the grid resolution and converges probably towards values suggested in Martínez-Tossas et al. (2016). The rotor torque is used as a global indicator for determining the distribution width, but the rotor thrust followed the

same trend. Hence it is interesting to see that while the rotor induction is predicted well, the velocity deficit agrees well only for $x/R > 5$ but not in the ultimate rotor vicinity.

It is also shown that with increasing grid resolution the spatial profiles seem to converge. This would be one aspect of a grid independent solution, but it is still very far away from resolving correctly the shed tip vortices. Although it seems to converge towards a value of $\epsilon/\Delta x \approx 4-5$ for which the dimensions of the experimental vortices would be attained, this causes excessive

computational costs due to the large mesh.

When looking at the turbulent inflow, it was shown that the decay of the turbulence intensity in longitudinal direction is much less pronounced than in previous work. As shown for the axial decay of the turbulence intensity a significant part of the difference between the resolved turbulence intensity and the imposed one from the synthetic field, resides within the sub-grid scales. Hence there is very little loss due to numerical dissipation which also reflects in the energy spectra which are the better

the higher the turbulent content is.

As expected the wake does recover at a faster pace for a higher turbulence intensity. It is very interesting to notice that the turbulent structures of the ambient flow eventually catch up with the amplitude of the structures emitted by the rotor. This is already noticable in the instantaneous velocity fields but becomes even clearer when evaluating the spectra. When considering the velocity fluctuations in the downstream flow caused by the blade passages for determining the near wake, it

can be observed that in this case for $TI_{syn} \geq 10\%$ the near wake already ends at $x/R = 0.4$. This is particular interesting as a turbulence intensity of $15\%$ at hub height is still considered to be low turbulence intensity according to ISO 61400 and many real sites exhibit even higher turbulence intensities. Hence for some cases the limit of the near wake would be $x/R = 0.4$ and even lower.

*Code and data availability.*   The SOWFA framework on which this work is based is made available by NREL https://github.com/NREL/

SOWFA/ and the turbulence generator for the homogeneous isotropic turbulence can be obtained via http://vbn.aau.dk/en/publications/
tugen(3e097a90-b3d8-11de-a179-000ea68e967b).html. The results for the NEW MEXICO experiments were provided upon request by Gerard Schepers.

*Competing interests.*   Christian Masson is a member of the editorial board of the journal.





*Acknowledgements.* This work is partially supported the Canadian Research Chair on the Nordic Environment Aerodynamics of Wind Turbines and the Natural Sciences and Engineering Research Council (NSERC) of Canada. Thanks for the great work done by Matthew Churchfield and colleagues at National Wind Technology Center, Boulder, CO, by establishing the open source framework SOWFA. The data used have been supplied by the consortium which carried out the EU FP5 project Mexico: 'Model rotor EXperiments In COntrolled

5   conditions'. Thanks also a lot to Gerard Schepers for providing results of the NEW MEXICO experiment.



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
