# Peer review of "Near wake analysis of actuator line method immersed in turbulent flow using large-eddy simulations"

_Wind Energy Science, 2018_

## Referee Comment (RC1) · Anonymous Referee #1 · 12 Apr 2018

General remarks:

This paper is of overall very good quality. A detailed discussion on the impact of the actuator line parameters is given. Even so no generic solutions are given, some metrics can be extracted from the work, regarding the optimal values of the Gaussian width parameter. Based on this validated actuator-line model, the authors propose a study on the impact of a homogeneous, isotropic turbulent inlet and shear layer turbulence. While the impact of the homogeneous isotropic turbulence is clear and well exposed thanks to the provided spectra, the additional impact of the shear layer is less obvious and should be discussed in more details. Specific comments: - P1 L7. Using a global

performance data may hide some error compensation. Why not using local forces or a more physical basis for Epsilon, such as the local chord?

- P3 L9. Airfoil polars as a function of the Reynolds number are not given in Schepers et al. Please clarify.

- P3 L12. The root correction by Shen is mentioned, but supposed to have "known outcome". Is it possible to clarify? Furthermore, other corrections have been proposed in the literature (see Snel, Chaviaropoulos, Bak, Dumitrescu. . .). Is it possible to include them?

- P3 L12. Coriolis / Centrifugal forcing enhance the lift of the airfoil near the root of the blade, not over the whole blade. It should be clearly stated here.

- P3 L12. It is disappointing to see that purely 2D airfoil polars are used, while 3D effects are discussed.

- P3 L16 → P4 L3. The authors argue the Glauert tip correction should be used due to the low resolution. According to Churchfield et al. (2017), this is due to the isotropic kernel that is used, and the virtual projection of forces outside of the "blade domain". It could be interesting to see which phenomena is dominating, i.e. the lower resolution or the isotropic projection. Furthermore, I was not able to find reference (Nathan, 2018). Is it already published? Otherwise, please mention it in the references.

- P6 L7. The domain is rather small in length compared with standard recommendations. As a comparison, N. Troldborg (2009) uses a domain length of 18R, Martinez-Tossas (2015) use a domain length of 21D. . . It could be useful to provide some proof of convergence.

- P6 L25. If possible, provide some orders of magnitude for the time step.

- P7 L6. The under-prediction of the axial induction is not clear to me; results are almost super-imposed to the NewMexico measurements.

- P8 L5 → L7. I do not understand the link with the actuator surface method. Even so epsilon over dx is adapted, depending on the Cartesian grid refinement, it is still an actuator line, and no chord-wise meshing is used.

- P8 L14 → 18. Discussion regarding the impact of the epsilon parameter is very instructive. Giving a look at the results, it seems to me there is an almost linear relation between the optimal epsilon value (leading to T/T_{ref} = 1) and the mesh refinement parameter N in the range 50 < N < 150. It could be interesting to derive an empirical law from it. In case the results presented in Figure 6 are not "rotor specific", this could lead to a very simple law to derive the value of epsilon "on the fly".

- P9 L13→L16. The "bad" resolution near the tip is, according to the authors, attributed to the actuator-line representation of the blades. In Blondel et al. 2017, a lifting-line model is used together with a vortex model of the wake, and better correlation with experimental data was obtained (compared to SOWFA simulations). Thus, from my point of view, discrepancies should be attributed to the isotropic kernel in use, which is unrealistic near the tip, or the potential excessive diffusion of the finite volume scheme. The effect of the isotropic Gaussian kernel and the mesh effects, as discussed P10., should be further analyzed (not necessarily in this publication, as this is not the main topic).

- P12. In this part, synthetic turbulence is imposed at the inlet of the domain. I guess results presented in Figure 11 are based on simulations without the wind turbines, considering only the evolution of the TI in a channel. A net decrease of the turbulence intensity is observed along the channel. The simulations are not really described here. It could be interesting to provide some proof of convergence. Is the simulation time long enough to transport the characteristics defined at the inlet? This could be a basic explanation to the decrease of the TI that is observed. Also, it seems something is happening at the outlet, with some kind of sharp TI recovery. If this belongs to ghost cells, it should not be present in this plot. The effect of the ratio between the CFD grid and the SEM grid is discussed. One can wonder about the turbulent length scales used

in the SEM algorithm, and their relation with the size of the computational grid. Is the CFD grid fine enough to catch the vortices given at the inlet?

- P14. Figure 15 is not useful and could be removed. Same remark holds for Figure 17, results are similar to the homogeneous isotropic turbulence case.

- P15. In 2.2.2., the evolution of mean velocity with height is presented. However, is seems to me, based on the contours of vorticity, that the TI is constant with height. Is that correct? From a physical point of view, higher TI is expected near the ground. This should lead to higher vorticity. Can the author provide some insight? In figure, y label should be z/R.

- P16. L7. I do not understand the point here. Please reformulate.

- P16. L11. Are the authors talking about the global rotor power? No metrics are given. The impact on vortex structures is not clear to me. Differences in the spectra between figure 19 and 14 are rather small. I would suggest including an additional metrics here to clarify the impact of the shear on the near wake.

- P17. Conclusions. Based on the observation of the turbulence decay (fig. 11) with axial position, it seems that at high TI, a large part of the TI is included in the subgrid scales. Therefore, I am not totally comfortable with the conclusions that are given: the length of the near-wake is determined based on the observation of the vorticity. However, the subgrid-scales are not included in the vorticity. Therefore, it seems difficult to draw definitive conclusions. As a more general remark, this work emphasize on the impact of the TI on the near-wake. However, blade loads may also be impacted, even at the airfoil polar level. One might expect a delay in the stall at high TI, which could impact the blade root loads. Also, it could have been interesting to use the NTNU Blind Comparison experiments as a complementary validation case.

Technical corrections:

- P1 L2. Noticable → Noticeable

- P1 L3. This works uses → This work uses

- P1 L24. To to assess → To assess

- P2 L12. "As done the former approach"

- P2 L29. Change "as the . . ." → With F representing. . . U the. . . (etc).

- P2 L32. Reformulate. "Finally, a summary is given. . ."

- P3 L8 Gaussian Kernel G → G the Gaussian Kernel

- P3 L8 Parenthesis: (f_{tip} → f_{tip}

- P3 L10 This data → These (plural)

- P4 L14. Latin abbreviations should be in italic (e.g.)

- P4 L15. Acronym HIT has never been defined

- P8 L6. .This can. . . (missing space)

- P8 L10. "here chosen method" (reformulate)

- P8 L16. E.g. → Italic

---

## Referee Comment (RC2) · Anonymous Referee #2 · 11 Jun 2018

The manuscript entitled "Near wake analysis of actuator line method immersed in turbulent flow using large-eddy simulations" deals with numerical computations of a modelled wind turbine embedded in LES computations. Several types of inflow turbulence are implemented in order to study its effect on the near wake development. The spectral content of the wake flow is then used as indicator of modifications. The present study is of interest for the wind energy community ad gives valuable insights on the near wake extent depending on turbulence intensities and types. On the other hand, the manuscript needs several improvements before to be accepted for publications.

- In general, the figure captions are poor and lack of essential information. The reader

should be able to understand the figure content just be reading the caption. Some axes are wrong. Some figures are not consistent with each other

- The manuscript should be self-consistent. Please remind the main parameters of the MEXICO / NEW MEXICO experiments: rotor dimensions, tip speed ratio, hub size, Reynolds number based on chord length, etc.

- What are the properties of the generated turbulence in term s of scales: integral length scale, ratio between this scale and the rotor radius? Since the authors present some turbulence spectra without wind turbines, it would be worth to develop a better description of the generated turbulence.

- The PhD thesis of the 1st author is cited regularly, whereas the reference is not precise enough to ensure that the reader will find it easily on the web. Additionally, this reference is sometimes cited for results which are not specifically a new outcome of this thesis. So please cite more relevant sources when possible.

Major comments:

- P2, lines 22-24: the spectra are also based on statistics. I do not see why it would be less sensitive than second-order statistics to the convergence issue, since the spectra is a frequency distribution of the variance.

- P3, line 10-12 : this data was obtained from wind tunnel experiments and therefore it does not include the stall delay due to boundary layer stabilizing effects such a Coriolis and centrifugal forcing which enhance the lift of the airfoil". These types of effects can be reproduced in a wind tunnel. Do you mean 2D experiment? Without rotation?

- Page 5, lines 4-6. Give some details about the synthetic turbulence fields: what are the turbulent length scales ? it seems that the turbulence does not dissipate with the axial distance. That is not so common, as explained later on in the manuscript. Some comments should be already mentioned here.

- Figure 5 : improve the caption and give details about the experimental configuration

used as reference here : power coefficient, thrust coefficient, TSR, etc

- Page 10, line 15-17 : this part is confusing : it seems that a relative decrease of turbulence is given (48% and 44%), whereas the 4% stands for a decrease of turbulence from 15% to 11%. This would correspond to a relative decrease of 25%... please rephrase this part.

- Page 16, lines 11-15. Please elaborate more on the discrepancies between sheared and un-sheared conditions

- Conclusion: there is not discussion about the relative size of the inflow turbulent structures compared to the wake turbulent structures (rotor size, blade size, tip vortex size, shear layer size?). It is indeed a very important parameter to justify the observations mentioned in page 17, lines 16-23.

Minor comments:

- P1, l24 : "to to"

- Figure 2 : if it the midplane at $y/R = 0$, the plot should be dependant of $z/R$ and $x/R$

- P3, line 8 : one parenthesis is missing

- P4, line 2 : "Kernel function"

- P4, line 3: please give the definition of sigma and Delta x.

- Figure 8 is too small. Additionally, it is difficult to differentiate the experimental and numerical results

- Page 10, line 2: The Reynolds number is based on the circulation: Please explain why you use this definition and not another one.

- Pge 10, line 7 : remove "here"

- Figure 10 : the caption is wrong

- Figure 11: the caption is poor and the authors should also better explain in the body text what this figure is for. Which computation solver is used here?

- Page 12, line 8 : ". . . should be at least twice as big" means dx/Delta x >2 ?.

- Figures 12 and 13 : Captions are not consistent with each other

- Figures 14 and 19: make both captions consistent. Spectra of what? Measured where? - Page 15, line 1: "determining the near wake" limit or boundary?

- Page 15, line 3: "reveals"

- Figure 18 : Y axis is not consistent with the caption

- Conclusion: remind in the conclusion the used method to generate the turbulent inflow

---

## Author Comment (AC1) · 30 Sep 2018

Dear Anonymous Referee #1,

thank you for your constructive comments and I hope to have them all addressed in a proper manner,

have a good day,

Jörn

PS: I am not 100% sure about the review process and I had a bit of a difficult time to understand from what I read on the website. I therefore submit my final response as

pdf and my latexdiff to have the difference between the current version and the one initially submitted.

Please also note the supplement to this comment:
https://www.wind-energ-sci-discuss.net/wes-2018-9/wes-2018-9-AC1-supplement.zip

---

## Author Response (AR2)

[revised manuscript text omitted]

**Anonymous Referee #1**

**General remarks:**

This paper is of overall very good quality. A detailed discussion on the impact of the actuator line parameters is given. Even so no generic solutions are given, some metrics can be extracted from the work, regarding the optimal values of the Gaussian width parameter. Based on this validated actuator-line model, the authors propose a study on the impact of a homogeneous, isotropic turbulent inlet and shear layer turbulence. While the impact of the homogeneous isotropic turbulence is clear and well exposed thanks to the provided spectra, the additional impact of the shear layer is less obvious and should be discussed in more details.

**Specific comments:**

- P3 L9. Airfoil polars as a function of the Reynolds number are not given in Schepers et al. Please clarify.

See P3, L9, Shen 2012 is taken as reference.

- P3 L12. The root correction by Shen is mentioned, but supposed to have "known outcome". Is it possible to clarify? Furthermore, other corrections have been proposed in the literature (see Snel, Chaviaropoulos, Bak, Dumitrescu. . .). Is it possible to include them?

At this point, it is difficult to include them. I removed controversal "known outcome".

- P3 L12. Coriolis / Centrifugal forcing enhance the lift of the airfoil near the root of the blade, not over the whole blade. It should be clearly stated here.

Done, see P3L11.

- P3 L12. It is disappointing to see that purely 2D airfoil polars are used, while 3D effects are discussed.

It was shown in Nathan 2017, that it is more than sufficient for the 15 m/s case.

- P3 L16 → P4 L3. The authors argue the Glauert tip correction should be used due to the low resolution. According to Churchfield et al. (2017), this is due to the isotropic kernel that is used, and the virtual projection of forces outside of the "blade domain". It could be interesting to see which phenomena is dominating, i.e. the lower resolution or the isotropic projection. Furthermore, I was not able to find reference (Nathan, 2018). Is it already published? Otherwise, please mention it in the references.

It would definitely be interesting, but it would be the matter of future work. Nathan, 2018 references to my PhD thesis, which should already be available since a while in a digital form at the library of ETS. I am in contact with the school in order to find out, why this has yet happened. I handed it in more than 6 months ago.

- P6 L7. The domain is rather small in length compared with standard recommendations. As a comparison, N. Troldborg (2009) uses a domain length of 18R, MartinezTossas (2015) use a domain length of 21D. . . It could be useful to provide some proof of convergence.

A complete sensitivity study was conducted in my PhD thesis. Including the numerous graphics and tables would blow up this article unnecessarily.

- P6 L25. If possible, provide some orders of magnitude for the time step.

Done, see P7L4.

- P7 L6. The under-prediction of the axial induction is not clear to me; results are almost super-imposed to the NewMexico measurements. C2

True, I rephrased that passage.

- P8 L5 → L7. I do not understand the link with the actuator surface method. Even so epsilon over dx is adapted, depending on the Cartesian grid refinement, it is still an actuator line, and no chord-wise meshing is used.

I agree, it is still an ALM, passage adapted.

- P8 L14 → 18. Discussion regarding the impact of the epsilon parameter is very instructive. Giving a look at the results, it seems to me there is an almost linear relation between the optimal epsilon value (leading to T/T_{ref} = 1) and the mesh refinement parameter N in the range 50 < N < 150. It could be interesting to derive an empirical law from it. In case the results presented in Figure 6 are not "rotor specific", this could lead to a very simple law to derive the value of epsilon "on the fly".

I share your enthusiasm on this. Probably this method should also be applied to e.g. NTNU blind comparison test or other experiments and see whether an empiricial relation could be derived. In my opinion it is slightly non-linear as seen in Fig.7.

- P9 L13→L16. The "bad" resolution near the tip is, according to the authors, attributed to the actuator-line representation of the blades. In Blondel et al. 2017, a lifting-line model is used together with a vortex model of the wake, and better correlation with experimental data was obtained (compared to SOWFA simulations). Thus, from my point of view, discrepancies should be attributed to the isotropic kernel in use, which is unrealistic near the tip, or the potential excessive diffusion of the finite volume scheme. The effect of the isotropic Gaussian kernel and the mesh effects, as discussed P10., should be further analyzed (not necessarily in this publication, as this is not the main topic).

I agree, I think it goes in hand with your point mentioned above and a possibility for future work.

- P12. In this part, synthetic turbulence is imposed at the inlet of the domain. I guess results presented in Figure 11 are based on simulations without the wind turbines, considering only the evolution of the TI in a channel. A net decrease of the turbulence intensity is observed along the channel. The simulations are not really described here. It could be interesting to provide some proof of convergence. Is the simulation time long enough to transport the characteristics defined at the inlet? This could be a basic explanation to the decrease of the TI that is observed. Also, it seems something is happening at the outlet, with some kind of sharp TI recovery. If this belongs to ghost cells, it should not be present in

this plot. The effect of the ratio between the CFD grid and the SEM grid is discussed. One can wonder about the turbulent length scales used C3 in the SEM algorithm, and their relation with the size of the computational grid. Is the CFD grid fine enough to catch the vortices given at the inlet?

Yes, the numerical grid cell size was chosen to respect the Nyquist-Shannon criterium, hence it was more than twice as refined as the synthetical grid.

- P14. Figure 15 is not useful and could be removed. Same remark holds for Figure 17, results are similar to the homogeneous isotropic turbulence case.

Done, removed.

- P15. In 2.2.2., the evolution of mean velocity with height is presented. However, is seems to me, based on the contours of vorticity, that the TI is constant with height. Is that correct? From a physical point of view, higher TI is expected near the ground. This should lead to higher vorticity. Can the author provide some insight? In figure, y label should be z/R.

Passage adapted, TI changes with height, but you are right, it is rather hard to notice with the vorticity plots.

- P16. L7. I do not understand the point here. Please reformulate.

Done, reformulated.

- P16. L11. Are the authors talking about the global rotor power? No metrics are given. The impact on vortex structures is not clear to me. Differences in the spectra between figure 19 and 14 are rather small. I would suggest including an additional metrics here to clarify the impact of the shear on the near wake.

Passage removed. It is discussed more in detail in my thesis, but it does not necessarily has to be in the article.

- P17. Conclusions. Based on the observation of the turbulence decay (fig. 11) with axial position, it seems that at high TI, a large part of the TI is included in the subgrid scales. Therefore, I am not totally comfortable with the conclusions that are given: the length of the near-wake is determined based on the observation of the vorticity. However, the subgrid-scales are not included in the vorticity. Therefore, it seems difficult to draw definitive conclusions. As a more general remark, this work emphasize on the impact of the TI on the near-wake. However, blade loads may also be impacted, even at the airfoil polar level. One might expect a delay in the stall at high TI, which could impact the blade root loads. Also, it could have been interesting to use the NTNU Blind Comparison experiments as a complementary validation case.

True, I think it is a shortcoming of my work, that I did not have the time to conduct this experiment with another rotor. It is true, that some of the turbulent motion falls within the subgrid, but as shown in my PhD thesis, more than 90% fall within the resolved scales. Hence I only take the resolved scales into consideration when looking at the limits of the near wake.

**Technical corrections:**

- P1 L2. Noticable → Noticeable C4

Done.

- P1 L3. This works uses → This work uses

Done.

- P1 L24. To to assess → To assess

Done.

- P2 L12. "As done the former approach"

Done.

- P2 L29. Change "as the . . ." → With F representing. . . U the. . . (etc).

Done.

- P2 L32. Reformulate. "Finally, a summary is given. . ."

Done.

- P3 L8 Gaussian Kernel G → G the Gaussian Kernel

Done.

- P3 L8 Parenthesis: (f_{tip} → f_{tip}

Done.

- P3 L10 This data → These (plural)

Done.

- P4 L14. Latin abbreviations should be in italic (e.g.)

Done.

- P4 L15. Acronym HIT has never been defined - P8 L6. .This can. . . (missing space)

Done.

- P8 L10. "here chosen method" (reformulate)

Done.

- P8 L16. E.g. → Italic

Done.

**Anonymous Referee #2**

The manuscript entitled "Near wake analysis of actuator line method immersed in turbulent flow using large-eddy simulations" deals with numerical computations of a modelled wind turbine embedded in LES computations. Several types of inflow turbulence are implemented in order to study its effect on the near wake development. The spectral content of the wake flow is then used as indicator of modifications. The present study is of interest for the wind energy community ad gives valuable insights on the near wake extent depending on turbulence intensities and types. On the other hand, the manuscript needs several improvements before to be accepted for publications.

- In general, the figure captions are poor and lack of essential information.
Discussion paper should be able to understand the figure content just be reading the caption. Some axes are wrong. Some figures are not consistent with each other

Several figure captions and figures were redone.

- The manuscript should be self-consistent. Please remind the main parameters of the MEXICO / NEW MEXICO experiments: rotor dimensions, tip speed ratio, hub size, Reynolds number based on chord length, etc.

Added, see P7L10-15.

- What are the properties of the generated turbulence in term s of scales: integral length scale, ratio between this scale and the rotor radius? Since the authors present some turbulence spectra without wind turbines, it would be worth to develop a better description of the generated turbulence.

Added length scale of incoming turbulence P5L9, R/4.

- The PhD thesis of the 1st author is cited regularly, whereas the reference is not precise enough to ensure that the reader will find it easily on the web. Additionally, this reference is sometimes cited for results which are not specifically a new outcome of this thesis. So please cite more relevant sources when possible.

Nathan, 2018 references to my PhD thesis, which should already be available since a while in a digital form at the library of ETS. I am in contact with the school in order to find out, why this has yet happened. I handed it in more than 6 months ago.

**Major comments:**

- P2, lines 22-24: the spectra are also based on statistics. I do not see why it would be less sensitive than second-order statistics to the convergence issue, since the spectra is a frequency distribution of the variance.

In wavenumber space it is easier to detect at which point the structures emitted from the blade merge with the surrounding turbulence. Also time series for energy spectra were taken at different points at the circonference of the rotor. Hence the spatial aspect can be neglected and it the energy spectra for each point can be averaged and the result is a smoother energy spectrum. This would not be possible with a radial profile of TI due to the asymmetric nature of the helicoidal vortical structures.

- P3, line 10-12 : this data was obtained from wind tunnel experiments and therefore it does not include the stall delay due to boundary layer stabilizing effects such a Coriolis and centrifugal forcing which enhance the lift of the airfoil". These types of effects can be reproduced in a wind tunnel. Do you mean 2D experiment? Without rotation?

Yes, passage was adapted.

- Page 5, lines 4-6. Give some details about the synthetic turbulence fields: what are the turbulent length scales ? it seems that the turbulence does not dissipate with the axial distance. That is not so common, as explained later on in the manuscript. Some comments should be already mentioned here.

Added, P5L9. It does dissipate, see Fig. 11, but at a lower pace than in other work.

- Figure 5 : improve the caption and give details about the experimental configuration used as reference here : power coefficient, thrust coefficient, TSR, etc

Done, P7L10-15.

- Page 10, line 15-17 : this part is confusing : it seems that a relative decrease of turbulence is given (48% and 44%), whereas the 4% stands for a decrease of turbulence from 15% to 11%. This would correspond to a relative decrease of 25%... please rephrase this part.

Done, rephrased.

- Page 16, lines 11-15. Please elaborate more on the discrepancies between sheared and un-sheared conditions

Removed passage.

- Conclusion: there is not discussion about the relative size of the inflow turbulent structures compared to the wake turbulent structures (rotor size, blade size, tip vortex size, shear layer size?). It is indeed a very important parameter to justify the observations mentioned in page 17, lines 16-23. Minor comments:

Done, P7L10-15.

- P1, l24 : "to to"

Done.

- Figure 2 : if it the midplane at y/R = 0, the plot should be dependant of z/R and x/R

Done.

- P3, line 8 : one parenthesis is missing - P4, line 2 : "Kernel function"

Done.

- P4, line 3: please give the definition of sigma and Delta x.

Done.

- Figure 8 is too small. Additionally, it is difficult to differentiate the experimental and numerical results

Done.

- Page 10, line 2: The Reynolds number is based on the circulation: Please explain why you use this definition and not another one.

Done, see P10L10-12.

- Pge 10, line 7 : remove "here"

Done.

- Figure 10 : the caption is wrong

Done.

- Figure 11: the caption is poor and the authors should also better explain in the body text what this figure is for. Which computation solver is used here?

Done.

- Page 12, line 8 : ". . . should be at least twice as big" means dx/Delta x >2 ?.

Done.

- Figures 12 and 13 : Captions are not consistent with each other - Figures 14 and 19: make both captions consistent. Spectra of what? Measured where?

Done.

- Page 15, line 1: "determining the near wake" limit or boundary? - Page 15, line 3: "reveals"

Done.

- Figure 18 : Y axis is not consistent with the caption

Done.

- Conclusion: remind in the conclusion the used method to generate the turbulent inflow

Done.